# Modelling the Decamerisation Cycle of PRDX1 and the Inhibition-like Effect on Its Peroxidase Activity

**DOI:** 10.3390/antiox12091707

**Published:** 2023-09-01

**Authors:** Christopher J. Barry, Ché S. Pillay, Johann M. Rohwer

**Affiliations:** 1Laboratory for Molecular Systems Biology, Department of Biochemistry, Stellenbosch University, Stellenbosch 7600, South Africa; cbarry@sun.ac.za; 2School of Life Sciences, University of KwaZulu-Natal, Pietermaritzburg 3201, South Africa; pillayc3@ukzn.ac.za

**Keywords:** enzyme kinetics, hydrogen peroxide, isothermal titration calorimetry, oligomerisation, parameter estimation, peroxiredoxin, quantitative redox biology, systems biology

## Abstract

Peroxiredoxins play central roles in the detoxification of reactive oxygen species and have been modelled across multiple organisms using a variety of kinetic methods. However, the peroxiredoxin dimer-to-decamer transition has been underappreciated in these studies despite the 100-fold difference in activity between these forms. This is due to the lack of available kinetics and a theoretical framework for modelling this process. Using published isothermal titration calorimetry data, we obtained association and dissociation rate constants of 0.050 µM^−4^·s^−1^ and 0.055 s^−1^, respectively, for the dimer–decamer transition of human PRDX1. We developed an approach that greatly reduces the number of reactions and species needed to model the peroxiredoxin decamer oxidation cycle. Using these data, we simulated horse radish peroxidase competition and NADPH-oxidation linked assays and found that the dimer–decamer transition had an inhibition-like effect on peroxidase activity. Further, we incorporated this dimer–decamer topology and kinetics into a published and validated in vivo model of PRDX2 in the erythrocyte and found that it almost perfectly reconciled experimental and simulated responses of PRDX2 oxidation state to hydrogen peroxide insult. By accounting for the dimer–decamer transition of peroxiredoxins, we were able to resolve several discrepancies between experimental data and available kinetic models.

## 1. Introduction

Peroxiredoxins (Prxs) are ubiquitous and highly abundant antioxidant proteins, which play crucial roles in reactive oxygen species detoxification, signalling, and heat stress response. The first member of the Prx family to be discovered was named “torin” after the toroid shape formed when Prx dimers oligomerise [1]; later, the peroxidase activity of Prxs was discovered and the family has been referred to as “peroxiredoxins” [2]. In addition to this oligomeric configuration, reduced and sulfenic Prx dimers are in equilibrium between the “fully folded” (FF) and “locally unfolded” (LU) conformations. In the FF conformation, peroxide substrates are able to bind in the active site and are thereby exposed to nucleophilic attack by the peroxidatic cysteine. By contrast, in the LU conformation, the peroxidatic cysteine is not available as a substrate and is instead brought closer to the resolving cysteine, thereby facilitating disulphide formation with the resolving cysteine [3,4]. Formation of the disulphide bond locks the Prx into the LU formation which destabilises the decamer [5]. The connection between Prx’s quaternary structure and peroxidase activity was established when it was demonstrated that abrogating decamer formation reduced the rate of hydrogen peroxide reduction by 100-fold [6]. However, this relationship has yet to be explored in any dynamic sense, likely owing to the lack of available kinetics or a theoretical framework.

The thermodynamic relationship between decameric and dimeric Prx has been described with $K_{d}$ values in the range of 1–2 µM^4^ or alternatively, since the interpretation of a $K_{d}$ cannot be directly linked to a concentration value for a fifth-order reaction, by a critical transition threshold (CTC) of approximately 0.8 µM, which defines the concentration above which all of the dimers would aggregate to form decamers [7], or by a C_0.5_ of 1.36 µM, which defines the concentration at which half of the total Prx protein, on a subunit molar basis, would be present in decameric form [8]. Curiously, a fully cooperative association of five dimers into a decamer cannot describe the “switch-like” relationship between total Prx and decameric Prx, where zero decamers are found below the CTC but above it they can be calculated as $Prx_{total}-CTC$. Instead, the equilibrium relationship has been described using mass action kinetics with Prx dimers raised to the power of 130 (instead of 5 which would be expected for a reaction with five reactants) and a dissociation constant of $2.4\times 10^{-10}$ µM^129^ [7]. At present, there is no mechanistic interpretation for this phenomenological description.

Given that both Prx decamers and dimers are present at equilibrium at concentrations relevant to in vitro assays, we can expect that these assays have measured the combined activities of both oligomeric forms. The $k_{\mathrm{cat}}$/$K_{M}$ of Prx during the reduction of hydrogen peroxide has been determined at 4–5 µM^−1^·s^−1^ with an NADPH linked enzyme assay [9] and at $10^{7}$–$10^{8}$ M^−1^·s^−1^ using horse radish peroxidase (HRP) competition assays [10,11]. The rate constants of obligate dimer and obligate decamer Prx mutants have been directly compared [6], but not the relative ratio of the rate constants of the different oligomeric forms of the wild-type. Indeed, even at 0.4 µM PRDX2, below the CTC where no Prx decamers are expected, a rate constant of $0.5\times 10^{8}$ M^−1^·s^−1^ in an HRP assay has been reported [11]. In the present study, we used in silico analyses to disentangle the effects of Prx dimers and decamers on in vitro peroxidase assays and to explore how Prx decamer activity might be observed at Prx concentrations below the CTC.

Most evidence suggests that the decameric form of Prx is dominant under reduced conditions [12,13] and thus, Prx oligomerisation has been excluded from computational models on the implicit assumption that Prx is always in the decameric form [14,15]. However, studying redox stress inherently requires the consideration of oxidised conditions. Simulations with a computational model of the Prx system in the red blood cell (RBC) using well-established kinetics [14] (termed “Model A” by the authors) showed Prx fully oxidised at hydrogen peroxide levels below 5 µM, which conflicts with in vivo studies showing only partial oxidation of Prx at this level of H_2_O_2_ [12]. In the model [14], this discrepancy was resolved by introducing an inhibited form of Prx, yielding “Model B”; however, this addition lacks a mechanistic explanation. Here, we investigated whether this discrepancy could, instead, be resolved by accounting for low-activity Prx dimers.

The question we aimed to address with this study was: can the Prx oligomerisation cycle be sufficiently described by reaction kinetics and incorporated into a kinetic model and, if so, what effect does this have on peroxidase activity? Incorporating the Prx decamerisation into a computational model required both the kinetics of the Prx dimer–decamer transition and a theoretical framework for modelling the peroxidase reactions of the 10-site Prx decamer. We were able to obtain the kinetics for the association and dissociation reactions by developing a model for isothermal titration calorimetry (ITC) and using it to analyse published ITC data [7]. We developed a new approach to modelling the activity of the Prx decamer that greatly reduced the complexity of the system, and show how best to incorporate these data into a kinetic model. Finally, we explore with kinetic modelling the influence of dynamically cycling Prx dimers and decamers on the in vitro peroxidase activity and the oxidation state of Prx in vivo. The cycling of Prx between the highly peroxidatically active decamers and less active dimers under hydrogen peroxide load has not been investigated before. Our results show that this process can be modelled and that it is crucial to understanding the role of Prxs in the cellular context as well as the relationship between Prx and redox homeostasis more broadly.

## 2. Materials and Methods

Modelling and computational analyses were performed in a Jupyter notebook [16,17] using the Python programming language [18] with the PySCeS [19], NumPy [20], SciPy [21], pandas [22], lmfit [23], sklearn [24], and Matplotlib [25] packages. The models, code and data for the simulations are available from https://github.com/Rohwer-Lab/Barry2023 (accessed on 25 August 2023); detailed instructions are provided in the repository’s README file and the required versions of the packages are specified in the requirements files. The models are described in detail in Appendix A and are available in both PySCeS input file (*.psc) format, as well as in the standard SBML (*.xml) format [26] from the GitHub repository. All analyses are available as Jupyter notebooks.

### 2.1. Topology of Decamerisation

Prx proteins undergo reversible concatenation from dimers to decamers (red square of Figure 1). Both dimers and decamers are able to undergo a redox reaction which converts H_2_O_2_ to H_2_O, although at different rates [6]. These oxidised Prxs undergo a condensation reaction by forming a disulphide bridge between the peroxidatic and resolving cysteines [27]. This disulphide bridge formation causes decamers to dissociate into dimers [5]. Additionally, Prx dimers follow the typical reaction network for Prxs, which includes sulfinilation [28] and subsequent reduction by sulfiredoxin [29], as well as reduction of disulphide Prx by the thioredoxin (Trx) system [30]. In this study, Prx dimers were modelled with two reaction sites that have the following oxidation states: reduced, SH; sulfenilated, SOH; disulphide bridge form, SS; or sulfinilated, SOOH. To unambiguously denote the different Prx dimer species, they were written such that the left-hand site followed the priority of SH > SOH > SS > SOOH (see also Section 3.5).

### 2.2. Model Construction

Kinetic models were drafted as a collection of stoichiometric and rate equations, parameters, and initial species concentrations, using the PySCeS input file syntax [19]. The models were formalised into sets of ordinary differential equations using the pysces.model() function and solved by the built-in PySCeS interface to CVODE [31]. See Appendix A for a summary and descriptions of the models used in this study.

### 2.3. Isothermal Titration Calorimetry Simulations

ITC experiment simulations were set up to mimic the experimental protocol followed by Barranco-Medina and co-workers [7]. To briefly recount their method, small volumes (1.6 µL) of a highly concentrated, and therefore decameric, Prx solution were repeatedly injected into a mixing cell, which initially contained a larger volume (1.4 mL) of buffer only. This rapid dilution causes the Prx decamers to dissociate into dimers in an endothermic process. After several injections, the concentration of Prx in the mixing cell increased to a value where the injected decamer no longer dissociated.

In this study, repeated injections were simulated with an interval matching the experimental protocol. In the model, these injection events were effectuated by an increase in the quantity of the various Prx species (depending on the model and $K_{d}$) and an increase in the reaction vessel volume.

The first injection of an ITC experiment is commonly ignored during data analysis [32] as diffusion between the contents of the reaction cell and the injection syringe causes reagents to partially react prior to the injection. However, the reagent still enters the reaction vessel and thereby influences subsequent injections. In our simulations, this was compensated for by initiating the system with a quantity of already dissociated Prx equal to that of a single injection.

The delay between the heat released by chemical dissociation and the instrument reading was accounted for using a Laplace transform [33]. Further details of the data processing procedure can be found in Appendix A.

ITC data [7] were digitised using the online data digitiser, WebPlotDigitizer [34]. For simulating the PRDX1 experiments, the following parameters were used: injection interval, 200 s; injection volume, 1.6 µL; total Prx dimers, 51 µM; PRDX1 dissociation enthalpy, 156 kcal/mol; and initial cell volume, 1400 µL.

### 2.4. Parameter Fitting

Parameters were estimated by non-linear least-squares regression of the digitised data to a model output of species concentrations from a simulation. The fitting was performed using the minimize() function of an lmfit.Minimizer() object with the “least-squares” method and an “epsfcn” parameter of 0.0001.

### 2.5. Simulations of the Prx in the Red Blood Cell

SBML code for the model of the Prx cycle in the human RBC was obtained from the authors of the original publication [14]. “Model B” was converted to the PySCeS (*.psc) input file format and modified as described below. To replicate the published simulations, a reaction for diffusion of hydrogen peroxide across the cell membrane, a process which was described in the original publication, was added to the model. To replicate “Model A” (the Prx cycle without inhibition [14]), all reactions involving the inhibited form of Prx, i.e., containing the species “iPrx_R_R_ox”, “iPrx_R_O_oox”, “iPrx_S_O2_disulf_form”, “iPrx_R_S_red_Trx1SH_Trx2SH”, “iPrx_R_S_red_Trx1SH_Trx2SOH”, and “iPrx_R_O2_srx”, were disabled.

The state of the Prx dimer population was described by several measures, as calculated below.

Fraction of Prx dimers without a disulphide bridge:(1)$$f_{dimers}\left(\mathrm{no}\;\mathrm{SS}\;\mathrm{sites}\right)=\frac{SH\_SH+SH\_SOH+SH\_SOOH+SOH\_SOH+SOH\_SOOH+SOOH\_SOOH}{\left[Total\;Prx_{dimers}\right]}$$

Fraction of Prx dimers with a single disulphide bridge only:(2)$$f_{dimers}\left(1\times \mathrm{SS}\;\mathrm{site}\right)=\frac{SH\_SS+SOH\_SS+SS\_SOOH}{\left[Total\;Prx_{dimers}\right]}$$

Fraction of Prx dimers with two disulphide bridges:(3)$$f_{dimers}\left(2\times \mathrm{SS}\;\mathrm{site}\right)=\frac{SS\_SS}{\left[Total\;Prx_{dimers}\right]}$$

Fraction of non-disulphide bridge Prx dimers with a sulfinic site:(4)$$f_{dimers}\left(\mathrm{SOOH}\;\mathrm{no}\;\mathrm{SS}\;\mathrm{site}\right)=\frac{SH\_SOOH+SOH\_SOOH+SOOH\_SOOH}{\left[Total\;Prx_{dimers}\right]}$$

Fraction of disulphide bridge Prx dimers with a sulfinic site:(5)$$f_{dimers}\left(\mathrm{SOOH}\;\mathrm{and}\;\mathrm{SS}\;\mathrm{site}\right)=\frac{SS\_SOOH}{\left[Total\;Prx_{dimers}\right]}$$

### 2.6. Horse Radish Peroxidase Competition Assay Simulations

The HRP competition assay measures the activity of Prx using competition with HRP for a substrate, in this case hydrogen peroxide. These assays are initiated using a stopped-flow apparatus to rapidly inject and mix the enzyme and substrate solutions. The course of the reaction is tracked by measuring the change in absorbance at 398 nm ($\Delta A_{398}$) originating from the absorbance of compound I, the product of the reaction of HRP with the substrate. Prx activity consumes hydrogen peroxide, which results in less compound I and a smaller $\Delta A_{398}$. The rate constant of Prx is calculated by comparing the $\Delta A_{398}$ in a reaction with Prx to one without Prx (refer to [10] for a detailed description) using the formula
(6)$$k_{Prx}\cdot Prx_{init}=k_{HRP}\cdot HRP_{init}\cdot \left(\left(\delta max-\delta obs\right)/\delta obs\right)$$
where $k_{Prx}$ and $k_{HRP}$ are the rate constants for Prx and HRP oxidation, respectively, $Prx_{init}$ and $HRP_{init}$ are the respective initial concentrations of Prx and HRP, δobs is the observed $\Delta A_{398}$, and δmax is the maximal $\Delta A_{398}$ found in the absence of added Prx.

In this study, HRP competition assays were simulated with a model containing the compound I, HRP (5 µM), Prx, hydrogen peroxide (1 µM) and a model without Prx as the control. The Δ*A*_398_ from the simulations was calculated as
(7)$$\Delta A_{398}=\epsilon_{398}\times \ell \times \Delta \left[\mathrm{compound}\;I\right]$$
where the extinction coefficient of compound I, $\epsilon_{398}=4.2\times 10^{4}$ M${}^{-1}\cdot$cm${}^{-1}$ [35], and a light path *ℓ* of 1 cm.

### 2.7. Time to Reach Equilibrium after Dilution Simulations

The dilution of the concentrated Prx was simulated in a PySCeS model by calculating the equilibrium concentrations prior to dilution, updating the model with these concentrations divided by the dilution factor, and then simulating the re-equilibration of the species over time.

### 2.8. Whole System Assay Simulations

Peroxidase assays were simulated in a model system that linked hydrogen peroxide reduction by Prx to NADPH oxidation, measured as decay at 340 nm, via Trx regeneration by thioredoxin reductase (TRR). The simulation parameters were as follows: Trx, 50 µM; TRR, 0.5 µM; $k_{cat,TRR}$, 10 s${}^{-1}$; hydrogen peroxide, 5 µM; NADPH, 150 µM.

## 3. Results

### 3.1. Modelling Prx with and without Decamer

In this study, Prx peroxidase activity was modelled in two ways. First, in what will be referred to as the mixed-activity dimer–decamer model, Prx decamer and dimer species were modelled explicitly using the published sulfenilation rate constant for decameric Prx and a value 100-fold lower for dimeric Prx [6]. Further, this model included a reaction for association and dissociation of the reduced decamer (red rectangle in Figure 1) as well as reactions for fast single-step dissociation of oxidised decamers once a single disulphide bridge was formed [5]. Second, in what will be referred to as the full-activity dimer-only model, Prx species were modelled explicitly as dimers only, except with the higher peroxidase activity as if they were perpetually in the decamer form. This is analogous to the topology of the green rectangle in Figure 1 with the Prx kinetics of the blue rectangle.

Kinetics for the dissociation of oxidised Prx were not available at the time of publication. However, decamer dissociation is driven by the formation of a disulphide bridge which locks dimers in the decamer-destabilising LU conformation [5]. Thus, we used a rate constant which ensured that the rate of dissociation of disulphide-bridge-containing decamers far exceeded that of other reactions in the system.

### 3.2. Models of Prx Decamerisation

Several approaches to modelling Prx oligomer association (the red rectangle of Figure 1) and peroxidase activity were considered and, ultimately, the transition was modelled as a single mass action reaction where five dimers associate into a decamer, using the published value of 1.1 µM^4^ for $K_{d(\mathrm{app})}$ [8].

We explored by simulation the distribution between dimers and decamers as a function of total Prx concentration (Figure 2). Dimers dominate at below the Prx CTC but decamers increasingly dominate above this, which is generally in line with the literature. However, the transition was not as stark as the experimental data, which found virtually no decamers below the CTC, and above it, the dimer concentration remained constant at the CTC [7].

### 3.3. Isothermal Titration Calorimetry Simulation

Determining kinetic constants requires time-based experimental data and, as far as we are aware, the only time-based (non-equilibrium) data available for Prx decamer dissociation are the ITC experiments of Barranco-Medina et al. [7]. Therefore, in order to characterise the association and dissociation constants of the Prx decamer, a kinetic model simulating the ITC experiments was developed. During development of the model, we used the exponent of 130 and a $K_{d(\mathrm{app})}$ of 2.4 × 10^−10^ µM^129^ as published in the original study to validate our approach (as discussed below, we subsequently used an exponent of 5 and a $K_{d(\mathrm{app})}$ of 1.1 µM^4^ when fitting the rate constants). With this model, ITC experiments with Prx were simulated allowing for the Prx species (Figure 3a) to be tracked and the rates (Figure 3b) of Prx decamer dissociation and association to be calculated over the time course of the experiment. Notably, the system behaviour changed around the 45 min mark, which corresponded to the ITC experiments of Barranco-Medina et al. [7] and the CTC. From the rates, the power (heat transferred per second) vs. time trace (Figure 3c) was calculated using:
(8)$$P=(rate_{decamer\;formation}-rate_{dimer\;formation})\cdot \Delta H$$
where *P* is the power and the known ΔH for Prx of 130–160 kcal/mol dimer was used in the simulations, depending on the Prx species [7]. The ΔH is often unknown when conducting an ITC experiment but can be determined from the *P* trace by finding the maximum area under the curve of all injections in an ITC experiment (Figure 3d).

Several of the experimental traces published by Barranco-Medina et al. [7] showed a single peak of heat generation at the CTC (the other peaks were endothermic), which the authors attributed to an association reaction. In the case of a system with multiple reactions at differing rates, an injection can produce an endothermic peak followed by an exothermic peak or vice versa; however, when this is observed, it occurs for several injections. The ITC experimental guidelines recommend using an injection period that allows the system to reach and rest at baseline between injections [36]. Failing to reach baseline produces a trace that superficially resembles baseline drift, another common issue associated with ITC experiments [37]. By considering the above, we were able to replicate the single peak following CTC observed by Barranco-Medina et al. [7] as an artefact by applying a baseline correction to an ITC trace of a system with an injection period that was not sufficient to reach baseline and had a sharp dissociation transition (Figure 3e with the area under the curve in Figure 3f).

### 3.4. Fitting $k_{on}$ and $k_{off}$

The Prx decamer association and dissociation rate constants where fitted using simulated ITC experiments and ITC time-course data of PRDX1 dissociation, digitised from Barranco-Medina et al. [7], and processed as detailed in Appendix A. The decamer dissociation and association reactions were described by the following mass action rate equations:(9)$$\begin{array}{c} v_{diss}=k_{off}\left[\mathrm{decamer}\right] \end{array}$$(10)$$\begin{array}{c} v_{ass}=k_{on}{\left[\mathrm{dimer}\right]}^{5} \end{array}$$
During parameter estimation, the value of $k_{on}$ was linked to $k_{off}$ (the fitted parameter) using the ratio
(11)$$k_{on}=\frac{k_{off}}{K_{d}}$$
and the published value of $K_{d}$ [8]. The best-fit values from this procedure were a $k_{off}$ of 0.055 ± 0.0011 (S.E.) s^−1^ and a $k_{on}$ of 0.050 ± 0.0010 (S.E.) µM^−4^·s^−1^ (Figure 4a). Until the CTC was reached, each simulated injection produced a similar heat release vs. time profile and the model consistently under-estimated the data except at approximately 20 s after the injection, where the heat release was, instead, overestimated (Figure 4b).

### 3.5. Enumerating the Molecular States of Decameric Prx

With the Prx decamer association and dissociation rate constants estimated, the final task required before incorporating decamerisation into a kinetic model was to describe the peroxidase-activity-associated reactions of the Prx decamer (the blue rectangle in Figure 1). Fully enumerated, there are $10^{4}$ possible configurations of the decamer (ten sites that can each exist in one of four states: fully reduced, sulfenic, sulfinic, or disulphide) and in the order of several million reactions; fortunately, several orders of magnitude of degeneracy can be introduced by using four assumptions.

First, mirror symmetry: since the two active sites of a Prx dimer are equivalent, the orientation of a dimer within a decamer can be rearranged by mirror symmetry (Figure 5a). This phenomenon is frequently applied in studies of protein oligomerisation and has already been used to reduce complexity when modelling Prx [14]. Second, rotational symmetry: the choice of “first position” in a linear representation of a circular molecule is arbitrary and can be selected as is convenient (Figure 5b). Third, planar symmetry: without planar distinctions, the plane of viewing a molecule can be changed as is convenient. In these three cases care must be taken that the relative positions of the dimers within the decamer remain the same and that the rate constants of each reaction are scaled by the corresponding number of degenerate reactions. Fourth, single-step decamer dissociation upon disulphide bridge formation: the intermediate Prx oligomer species constitute a small fraction of the overall Prx population and the overall effect of including them is likely to be negligible; therefore, oxidised decamer dissociation was assumed to take place in a single step.

Using the above assumptions, the peroxidase-activity-associated reactions of the Prx decamer were enumerated using a Python script, without the sulfinilation reactions. Following this, we drafted dissociation reactions for each decamer, enumerated by the algorithm, that contained an SS site. In total, this reduced the model complexity to describe Prx decamer oxidation and dissociation, so that in the end only 270 reactions and 133 species were needed. The logic of the reaction enumerating script is further described in Appendix A.

We found that, when including the sulfinilation reactions in a model, the number of reactions was computationally prohibitive. No kinetics are available for the difference between the sulfinilation rate constant of the dimer and that of the decamer. Considering that disulphide bridge formation is likely the primary driver of decamer dissociation, the proportion of decamers that contain hyperoxidised sites during the catalytic cycle is likely to be low. Therefore, the Prx decamer reaction network without sulfinilation reactions was incorporated into a kinetic model of the Prx system.

### 3.6. Comparing the Prx Red Blood Cell Model by Benfeitas et al. [14] with and without Decamerisation

Neither the kinetics for PRDX2 decamer association and dissociation nor a validated in vivo model of PRDX1 was available at present; thus, we added the PRDX1 kinetics to a model of PRDX2 for the RBC. The two Prx isoforms bear 77% sequence identity [38] and have similar sulfenilation rate constants [39], with the most relevant distinction being that the rate of disulfide formation for PRDX1 is ∼55× that of PRDX2 [39], diverting Prx away from the sulfinilation reaction. Considering that sulfinilation had a minor influence on simulations (Figure 6), we concluded that PRDX1 was a suitable stand-in for PRDX2 in this instance.

The effect of Prx decamer association and dissociation on the molecular state of the Prx population (as might be observed in SDS-PAGE and Western blot studies) was evaluated by adding the Prx dimer–decamer transition to a previously published independent-sites model of PRDX2 peroxidase activity. Incorporating this process included adding a reaction for association and dissociation of the reduced Prx decamer, peroxidase reactions of the Prx decamer, fast single-step dissociation reactions for oxidised decamers once a single disulphide bridge was formed, as well as setting the Prx decamer sulfenilation rate constant to that published with the original model and assigning a rate constant 100-fold lower for the dimeric Prx [6]. Model A [14] contains the Prx cycle based solely on the available kinetics without allosteric interactions between the subunits of a dimer (i.e., independent dimer sites), and predicts that Prx consumes the overwhelming majority of hydrogen peroxide at low hydrogen peroxide concentrations. In that work, the authors added an inhibited form of Prx to replicate the experimental evidence, which shows equivalent contributions of catalase and Prx to hydrogen peroxide consumption at low hydrogen peroxide concentrations [40,41,42,43]; this was termed Model B [14]. Here, we added the Prx decamer association and dissociation reactions with kinetics as determined (Figure 4) to Model A (termed Model A with decamerisation). The steady-state response of the fraction of Prx dimers to changes in hydrogen peroxide concentration of Model A with decamerisation (Figure 6a) was able to reproduce the results of Model B (Figure 6e) *without the addition of an inhibited form* of Prx. This behaviour was quite different from the original Model A (Figure 6c). A similar result was found for the time-dependent response to a hydrogen peroxide bolus (Figure 6b,d,f). By introducing decamerisation into Model A, we were therefore able to approximate the experimental data without the further introduction of an inhibited form of Prx, for which there is no direct evidence (see Discussion). Note that, whereas the Prx oxidation state in Model B could be reproduced by Model A with decamerisation, the same was not true for the contributions to hydrogen peroxide consumption in the resting state. The $v_{Prx}:v_{catalase}$ at low hydrogen peroxide for Model A with decamerisation was just marginally reduced (387.6 compared to a value of 388.1 for Model A), and the value was much higher than the 1:1 observed for Model B. Further comparisons between Model A and Model A with decamerisation regarding hydrogen peroxide metabolism and other redox factors are provided in Appendix A as reproductions of the analyses found in Benfeitas et al. [14].

### 3.7. Incorporating Prx Decamers Lowers Activity

To determine the extent to which Prx decamer dissociation influences Prx activity, we simulated assays comparing the full-activity dimer-only model to the mixed-activity dimer–decamer model (see Section 3.1 for definitions of these models). Comparing the same-colour traces of the simulated HRP competition assay in Figure 7a shows that including dimers with lower peroxidase activity and the dimer–decamer equilibrium resulted in an overall decrease in the peroxidase activity of Prx, indicated by a larger $\Delta A_{398}$ (a larger $\Delta A_{398}$ indicates that less hydrogen peroxide was consumed by Prx). Analysing the fractional inhibition of the simulated traces (Figure 7b) yielded rate constants of 175 µM^−1^·s^−1^ and 91 µM^−1^·s^−1^ for the full-activity dimer-only model and the mixed-activity dimer–decamer model, respectively.

Next, to evaluate this decrease in activity in a dynamic system where Prx is regenerated after oxidation, we simulated an NAPDH oxidation assay with Prx, Trx, and TRR (Figure 7c). Similar to the simulated HRP assay, including low-activity dimers in the model lowered the Prx peroxidase activity, indicated by a shallower initial slope (a smaller ΔNADPH indicates that less NADPH has been oxidised and, therefore, less hydrogen peroxide has been consumed by Prx). The rate constants for the full-activity dimer-only model and the mixed-activity dimer–decamer model were determined by initial rate kinetics to be 2.71 s^−1^ and 2.09 s^−1^, respectively (Figure 7d). From these simulations, it is clear that models that partition the reduced Prx population into high-activity decameric Prx and low-activity dimeric Prx, exhibit lower overall Prx activity than models that only consider high-activity Prx. This effect is inversely correlated to the Prx concentration since, although both Prx dimers and decamers increase with total Prx pool size, dimers occupy a relatively smaller proportion of the total Prx pool as the Prx concentration increases. Although this inverse relationship is not as marked as when modelling the Prx dimer–decamer equilibrium stringently according to the CTC observed by Barranco-Medina et al. [7], it provides general support for such a dimer–decamer transition.

To investigate possible reasons for the reduction in peroxidase activity when incorporating the dimer–decamer transition, we calculated the disequilibrium ratio ($Q/K_{d}$, where *Q* is the reaction quotient) for the equilibrium between reduced dimers and decamers (Appendix A). While the system was under hydrogen peroxide stress, the proportion of the reduced dimeric Prx pool was greater than that of the same system at rest and the reaction was out of equilibrium ($Q/K_{d}>1$, see Appendix A). Furthermore, we calculated the fraction of reduced sites in fully reduced and hetero-oxidised dimers. The hetero-oxidised dimers are unable to form decamers and, therefore, the reduced sites contained in them will have the corresponding low peroxidase activity of dimeric Prx. In simulated HRP and NADPH assays, this fraction of hetero-oxidised dimers increased upon hydrogen peroxide exposure (Appendix A).

We also evaluated whether our results depended on the assumption that decamers dissociated immediately upon formation of a single disulphide bridge (Appendix A). Expanding the model to allow for the formation of up to one disulphide bridge on each dimer of the decamer (i.e., a maximum of five disulphide bridges in total), and to include decamer dissociation reactions for all the intermediate species, did not substantially alter the model behaviour: simulations of the RBC and NADPH assay produced identical results, while the inhibition of peroxidase activity in HRP assays was slightly attenuated (Appendix A). The model size was increased significantly (e.g., a total of 653 species and 2321 reactions for the HRP assay model), but was nevertheless tractable. However, allowing for two disulphide bridges per dimer unit in the decamer led to a combinatorial explosion of species and reactions that could no longer be feasibly modelled.

### 3.8. Diluting Prx Can Influence Peroxidase Activity

We were curious whether decamer dissociation following dilution of a Prx solution whose concentration is above the CTC needs to be considered by researchers handling Prx in the laboratory. Specifically, we considered dilution of frozen protein solutions where, commonly, purified Prx solutions are stored at approximately 10 mg/mL [44,45,46], which translates to ±450 µM (using an Mr of 21.892 kDa for PRDX2 [47] via Uniprot), as well as dilutions of 10–100 µM Prx as would occur during HRP competition assays. Our simulations show that following a 10× (Figure 8a) or 50× (Figure 8b) dilution, solutions ranging in concentration between 1 and 450 µM Prx equilibrated over a time span of ±90 s to as fast as <1 s. Comparing Figure 8a to Figure 8b, we find that increasing the magnitude of the dilution resulted in a longer time to equilibrium for all Prx concentrations except for the 1 µM solution, which took the same time to reach equilibrium following a 10× or a 50× dilution. Additional examples of solution dilution simulations can be found in Appendix A.

This equilibration is fast enough that a Prx solution would reach equilibrium after dilution in most laboratory applications, with the exception of HRP competition assays of the peroxidase activity of Prx, which can run to completion in approximately 100 ms [10,11,44]. To evaluate the extent that this affects assays of Prx peroxidase activity, we simulated an HRP competition assay (Figure 8c) using the mixed-activity dimer–decamer model and compared the hypothetical scenario of Prx dimers and decamers equilibrating instantly to the scenario of Prx dimers and decamers equilibrating as per their mass action kinetics following injection by a stopped-flow apparatus, i.e., dilution at t=0. These traces show that an instantly equilibrated system is expected to have lower peroxidase activity than one that follows the dissociation kinetics, which is confirmed by the rate constants of 167 µM^−1^·s^−1^ and 190 µM^−1^·s^−1^ (Figure 8d), obtained from simulations with pre-equilibrated Prx and from those following the dissociation kinetics after dilution, respectively. Together, these results show that the equilibration of Prx following dilution can affect HRP competition assays when the time between dilution and assay is sufficiently short that the system cannot reach equilibrium, but should not affect most other assays, where there is a longer time between dilution and data acquisition.

## 4. Discussion

The capacity of Prx to form reduced decamers and then to dissociate into dimers upon oxidation is well established [48]; however, to date, no studies have incorporated this process into kinetic models. In part, this is due to the difficulty in measuring oligomerisation kinetics and researchers have circumvented the issue by modelling Prx activity with dimer topology and decamer kinetics [14,15], i.e., with the assumption that Prx is always in the high-activity decamer form. Evidence supports this assumption under basal conditions and low hydrogen peroxide [12,13]; however, it is desirable to model the response of cellular redox factors and hydrogen peroxide protection mechanisms under hydrogen peroxide load to understand how these systems respond to oxidative stress. We endeavoured to model the dimer–decamer transition of Prx in a dynamically responding system of Prx activity.

We were able to, for the first time for any Prx, determine the association and dissociation rate constants for the dimer–decamer transition as 0.050 µM^−4^·s^−1^ and 0.055 s^−1^, respectively. Using the tools of computational biology we were able to extract additional value from data that have been publicly available for over a decade [7] by performing a relatively complex analysis to derive these novel kinetic parameters. By measuring heat release directly, ITC allows for measurement of oligomerisation reactions without fluorogenic prosthetic groups and has been used extensively to study physical and chemical binding equilibria and to determine thermodynamic parameters in molecular biology [49]. Although the application of ITC to enzyme kinetics has long been established [50], it has not been as widely adopted as other kinetic assay techniques, owing to several factors that complicate data analysis. An upper limit of 2 s^−1^ has been proposed for determining kinetic parameters from ITC [51], which is well above the values determined in this study. This gives us a high degree of confidence in the the kinetics presented here; however, as with any first time reporting, confidence in the accuracy of the Prx oligomerisation kinetics would be greatly improved by additional studies, in particular ones with additional experimental data. Our work highlights the power of ITC as an aid for the construction of systems-biology models, as time-dependent rate constants are crucial to simulating dynamic behaviour.

With these kinetic parameters, we were able to model the reduced Prx pool as a dynamic population of decamers and less peroxidatically active dimers. Comparing this to a model of the Prx pool using the topology of dimers with the kinetics of decamers (Figure 7), we found that our expansion of the model resulted in less peroxidase activity, an effect that arose from three factors: first, in a system at rest, the equilibrium between Prx dimers and decamers requires that a portion of the reduced Prx pool is dimeric, roughly corresponding to the CTC of 0.8 µM [7]. Second, since dimers contain two active sites, a reduced Prx site will regularly be paired with a site that is either sulfenic, disulphide, sulfinic, or—although not modelled in this study—sulfonic during Prx oxidation and regeneration. These hetero-oxidised dimers are unable to form decamers and, therefore, the reduced sites contained in them will have the corresponding low peroxidase activity of dimeric Prx (Appendix A). Third, when the sum of the rates of Prx regeneration and decamer dissociation is greater than the rate of decamer formation, the pool of reduced dimers will accumulate until the net rate is zero and returns to equilibrium. In this way, while a system is under hydrogen peroxide stress, the proportion of the reduced Prx pool that is dimeric will be greater than that of the same system at rest (Appendix A).

We adopted a systems biology approach and evaluated the effect of the Prx dimer–decamer transition on the Prx redox cycle in silico by expanding an existing model of PRDX2 activity [14]: we added the reduced Prx decamer association and dissociation reactions (with the associated kinetics determined in this study), dissociation of the oxidised decamer, and included a 100-fold lower activity for the Prx dimer than the decamer. Simulations of Prx species oxidation vs. hydrogen peroxide supply (Figure 6) show good agreement with the SDS-PAGE results of the same system [12]. Considering that the parameters used to model the Prx dimer–decamer transition are derived entirely independently, we view this as a strong validation that the Prx dimer–decamer transition is essential to modelling the dynamics of the Prx redox cycle. Furthermore, the RBC contains far above the average level of Prx, which across various tissues is in the low dozen micro-molar range as derived from proteomics data [52]. Hence, the Prx dimer should have a proportionately larger effect on Prx activity and the oxidised species profile in other cell types compared to the RBC.

During construction of their erythrocyte PRDX2 independent-site model with no allosteric interactions between subunits in a dimer [14] (termed “Model A”), Benfeitas and co-workers found that there was a large discrepancy between their simulations and the aforementioned SDS-PAGE experimental evidence [12]. To reconcile these discrepancies, they augmented Model A with a hypothetical inhibition of Prx to create Model B while granting that there was no evidence for this inhibitor. Based on the evidence that Prx and catalase contribute comparably to peroxidase activity at low hydrogen peroxide [40,41,43], the authors parameterised the inhibitor such that it had near equilibrium binding and disabled the peroxidatic activity of >99% of Prx. This sequestration-based inhibition bears similarity to our approach in that reduced Prx sites can be “sequestered” into a low-activity dimer until they can reform into a decamer but differs in that the entire Prx population is immediately available in the event of a surge in hydrogen peroxide. Without using kinetics derived in any part from their Model B, our simulation of Model A with the Prx dimer–decamer transition added was able to replicate the response of PRDX2 oxidation state in Model B almost perfectly in both steady-state and hydrogen peroxide bolus simulations. Therefore, we propose that the Prx dimer–decamer transition with lower peroxidase activity of the Prx dimer is, in fact, the mechanism by which the discrepancy in the PRDX2 oxidation state between model [14] and the data [12] can be resolved.

In our approach we modelled the process of Prx decamer association and dissociation as a single reaction instead of a series of concatenating or splitting steps. Implicitly, this assumes perfect cooperativity and negligible presence of the intermediate oligomers, which is congruent with several studies [7,8]. However, it could be argued that while these assumptions are valid given the current limits of available techniques, this may not hold up to methods with greater resolution. Examples which tenuously hint that relevant amounts of intermediate Prx oligomers may be present in low concentrations of reduced Prx include the gradual increase in ΔH for the initial injections of several of Barranco-Medina’s ITC experiments [7] as well as the slight non-linearity visible in Villar’s phasor plots [8]. Therefore, the validity of the assumption of perfect cooperativity will need to be reevaluated when more evidence becomes available.

Assigning a dissociation scheme to a particular Prx isoform is complicated by the lack of detailed mechanistic and kinetic studies of the process. This complexity is heightened by the presence of inter-isoform variation [53], suggesting that inter-species variation is equally plausible. The Prx decamer undergoes destabilisation when dimers adopt the LU configuration, and disulphide bridge formation traps the dimer in this conformation [5]. Consequently, our modelling approach included dissociation of oxidised decamers upon formation of a single disulphide bridge. This strategy aligns with previous findings, albeit involving mutants, that reported fully reduced dimers after subjecting reduced decamers to hydrogen peroxide [54]. However, a contrasting observation has been documented, wherein fully disulphide bridged decamers, featuring all constituent sites in the disulphide bridge state, could be observed in an equilibrium with dimers with a reported $K_{d}$ of ≈10 µM^4^ [11]. In that same study, fully reduced dimers could not be resolved, suggesting a stronger association of reduced decamers, which is consistent with the lower $K_{d}$ of 1.1 µM^4^ from a later report by the same group [8]. The latter value was also used as a basis for our model.

Our initial computational model assumed a simplified, one-step, irreversible decamer disassembly process upon the formation of a single disulphide bridge, with the disulphide-bridge-containing decamer being insensitive to sulfenilation. This adaptation was a trade-off for computational feasibility, balanced by the rapid rate of oxidised decamer dissociation. Nevertheless, since the existence of decamers with multiple disulphide bridges has been reported as discussed above, we verified that a more elaborate reaction scheme including these species would not substantially change our findings and conclusions (compare Appendix A to Figure 6 and Figure 7). By way of explanation, the system is either entirely decameric (above the CTC) during faster HRP assays or primarily dimeric during extended assays, as seen in RBC simulations, or when concentrations are too low for decamers to form, as observed in the NADPH assay. We did observe a slightly higher peroxidase activity for the extended model in the HRP assay, where the inclusion of sulfenilation after disulphide bridge formation narrowed the activity gap between the full-activity dimer-only model and the mixed-activity dimer–decamer model. Overall, under high hydrogen peroxide load, decamer dissociation has minimal impact but dimer to decamer association becomes limiting. Incorporating the dimer–decamer transition is, therefore, pivotal for understanding both the Prx oxidation state response to hydrogen peroxide insult and Prx peroxidase activity. Nonetheless, the implications of Prx decamer–dimer dissociation for Prx activity and the oxidation profile response to hydrogen peroxide require ongoing scrutiny, particularly with further elucidation of the dynamics and kinetics of the process.

Our study has a number of additional limitations: there are only a few reports on the difference in oxidation kinetics between the dimeric and decameric forms of Prx; there is a complete lack of kinetics for the dissociation of oxidised Prx; and the simulation of large models comprising thousands of reactions comes with a significant computational strain. The influence of the Prx dimer–decamer transition on the activity of a particular Prx species is entirely dependent on a difference in activity between the dimeric and decameric forms. We hope that our findings will stimulate interest in this field and that this aspect will be investigated for a wider range of Prx species. With regard to the kinetics for dissociation of oxidised Prx, sulfinilation reactions of the decamer were excluded. However, if this process is found to proceed at a rate comparable to sulfenilation or disulphide bridge formation, it will be necessary to include the reactions for sulfinilation of the Prx decamer as well. Additionally, incorporating each of these processes would exponentially increase the size of the model, most likely beyond the capabilities of personal computers, and may require high-performance analysis on a computing cluster if at all computationally tractable.

If Prx dimers have lower Prx activity than Prx decamers, should we not consistently find that the Prx activity drops off around the CTC of the Prx decamer in peroxidase assays? Indeed, we believe that evidence for this can be found in several studies but, to our knowledge, has thus far been overlooked. The studies of Prx activity by HRP competition assays [10,55,56,57,58] show that a linear fit of fractional inhibition vs. Prx has an x-axis intercept in the 0–0.5 µM range, suggesting that there is ∼0–0.5 µM of peroxidatically inhibited Prx. As a counter-example, Manta and co-workers [11] report a peroxidase rate constant of $0.5\times 10^{8}$ M^−1^·s^−1^ for 0.4 µM PRDX2, and while this is lower than the rate constant they reported for higher Prx concentrations (1.0, 1.2, and $1.1\times 10^{8}$ M^−1^·s^−1^ for 0.8, 1.3, and 1.7 µM Prx, respectively), it is not congruent with a CTC of 0.8 µM [7] and a 100-fold lower Prx activity for dimers than for decamers [6] (see also discussion below).

As a caveat, the rate constants derived by analysing simulations of HRP assays reported here should only be considered in comparison to each other and not as kinetics for PRDX1. The established fractional inhibition method for determining rate constants from HRP competitive assays systemically and increasingly misestimates the target rate constant, the more this target rate constant differs from $k_{HRP}$ [59].

The observed rate of hydrogen peroxide consumption during peroxidase activity assay with Prxs is a combination of the activity of the dimeric and the decameric Prx. This raises the question of how published Prx rate constants should be interpreted. Considering that, typically, Prx is incubated with DTT prior to activity assays to ensure that it is fully reduced [10] and, canonically, reduced Prx is decameric, this has lead to the equating of reported Prx rate constants to decameric Prx rate constants, which has carried through implicitly to modelling studies [14]. Since the Prx population is divided between dimers and decamers, decameric Prx is likely overestimated in these assays. Considering that catalytic rate constants are determined by dividing activity by enzyme concentration, this leads us to the conclusion that reported rate constants for decameric Prx are somewhat under-estimated, which is supported by the finding that obligate decamer mutants of Prx have higher activity than wild-type [6]. Confoundingly, this effect has been masked in millisecond assays by the fact that Prx is unable to fully dissociate prior to oxidation, since these assays are initiated by diluting the enzyme solution by mixing with the substrate solution (Figure 8). This may explain why Manta and co-workers [11] were able to detect significant peroxidase activity at concentrations of PRDX2 where only low-activity dimers should have been present. Protein concentrations prior to substrate addition and injection volumes or assay-to-stock dilution ratios are rarely reported, making it difficult to evaluate the extent of these two effects on published Prx kinetics, an issue that could be alleviated by improved assay reporting standards [60,61,62].

## 5. Conclusions

In this work, we have developed a model of the Prx decamer association and dissociation cycle. This model was parameterised by fitting the relevant rate constants to digitised ITC data for Prx dilution experiments. We developed a script that enumerates the reactions of the Prx decamer oxidation cycle. Together, these reactions were incorporated into an established model of hydrogen peroxide neutralisation by PRDX2 in the RBC. With this, we demonstrated that incorporating decamer dissociation causes an inhibition-like effect on peroxidase activity. This allowed us to mechanistically resolve a discrepancy between experimental data and kinetic simulations by showing that reduced Prx sites can be sequestered in a less active dimeric form. Additionally, we have demonstrated that Prx decamer dissociation occurs within a time frame relevant to peroxidase assays and other oxidation experiments and needs to be considered when working with Prx in a laboratory. Our findings strongly emphasise the link between the quaternary structure of Prxs and their peroxidase activity, a connection that has been underappreciated so far.

In conclusion, Prx kinetics have been studied using a range of methods such as steady-state and competition kinetic assays, ITC as well as newer methods such as phasor analysis. Computational modelling offers a platform to combine and organise different experimental data into a single framework to better understand these important antioxidant proteins.

## Figures and Tables

**Figure 1 antioxidants-12-01707-f001:**
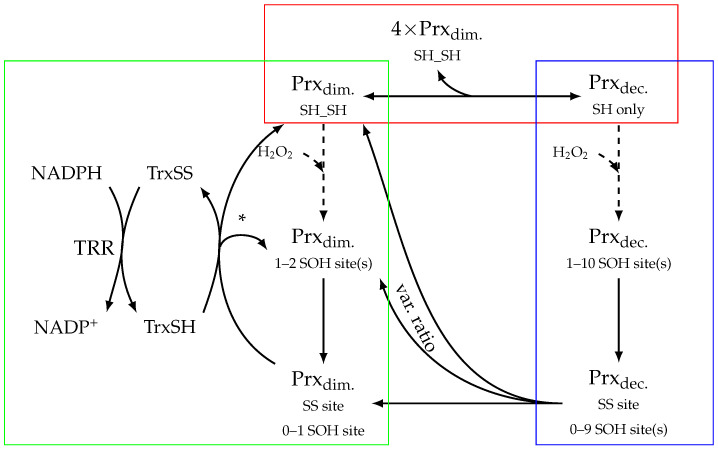
Peroxiredoxin (Prx) decamer association and dissociation pathways and their relationship to 2-Cys Prx activity and regeneration. Decamerisation can be modelled as a reaction where five dimers associate into a decamer in a single step. Prx sites are denoted as follows: SH, reduced; SOH, sulfenilated; or SS, disulphide bridge. SS sites are regenerated by thioredoxin (Trx), which is, in turn, regenerated by thioredoxin reductase (TRR) using NADPH. Red rectangle: the reactions involved in decamer association and dissociation. Blue rectangle: the reactions of Prx decamer peroxidase activity and disulphide bridge formation. Sulfinilation and reduction by sulfiredoxin are omitted for brevity. Green rectangle: the reactions of Prx dimer peroxidase activity, disulphide bridge formation, and Prx regeneration. Dashed arrows represent reactions that can occur recursively. Prx_dim._ and Prx_dec._ are placeholders for several Prx dimer species and the multitude of Prx decamer species, respectively, with the number of chemical residues of each oxidation state listed below. * One Prx dimer species is formed per reaction.

**Figure 2 antioxidants-12-01707-f002:**
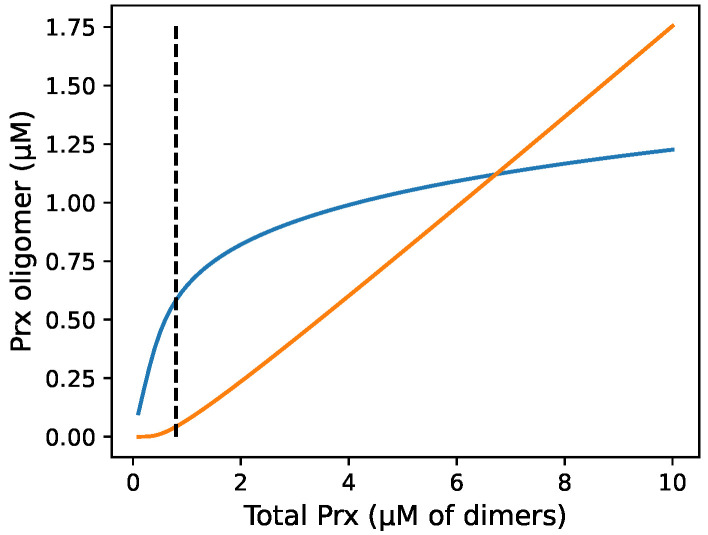
The proportion of reduced Prx oligomers that are in decameric or in dimeric form at equilibrium is related to the total reduced Prx concentration. The figure shows an equilibrium simulation of the distribution of dimeric and decameric PRDX1 over a range of total PRDX1 concentrations using a one-step mass action model with an exponent of 5. (**—**) Prx dimers, (**—**) Prx decamers, and (**- -**) the critical transition threshold or minimum concentration of total Prx where decamers are observed as described by Barranco-Medina et al. [7].

**Figure 3 antioxidants-12-01707-f003:**
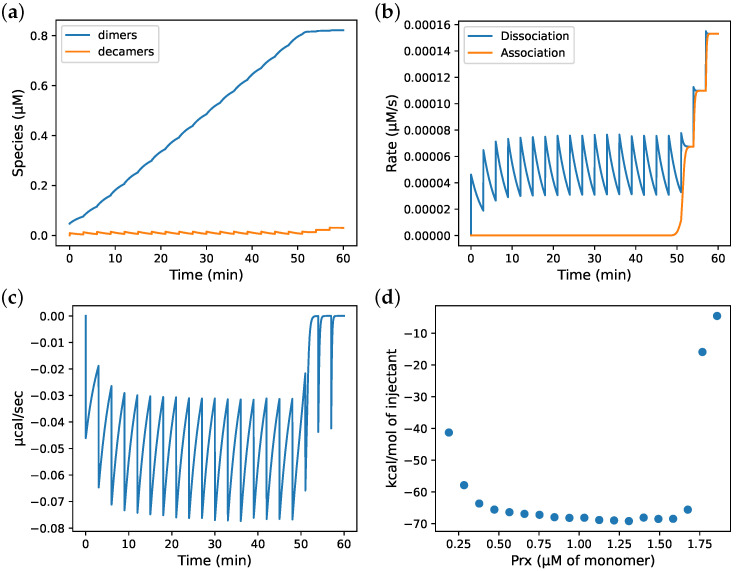
Development of a model for simulating isothermal titration calorimetry (ITC) experiments. The rates and species concentrations, as well as heat generation, were calculated over time with model simulations. (**a**) Trace of species concentration over time; (**b**) rates over time; (**c**) heat generation over time calculated by the product of association enthalpy and the difference between dissociation and association rates; (**d**) heat released by each injection calculated as the area under the curve; (**e**) heat generation over time calculated as in (**c**) and with the baseline subtracted; (**f**) heat released by each injection calculated as the area under the curve after baseline correction. The injection parameters were set to match those in Figure 1A of [7] where the top panel is comparable to (**c**) and (**e**) and the bottom panel comparable to (**d**) and (**f**). A full injection was assumed to occur before the beginning of the trace (the first injection of an ITC experiment does not produce a full heat curve but must be considered in the model simulation). Model parameters were as follows: reaction exponent, 130; $K_{d(\mathrm{app})}$, 2.4 × 10^−10^ µM^129^; $k_{on}$, 0.0021 × 10^10^ µM^−129^·s^−1^; $k_{off}$, 0.005 s^−1^; injection volume, 1.6 µL; syringe total Prx (in dimers), 87.5 µM; injection interval, 180 s; association enthalpy, 142 kcal/mol of Prx dimer (*Arabidopsis thaliana*).

**Figure 4 antioxidants-12-01707-f004:**
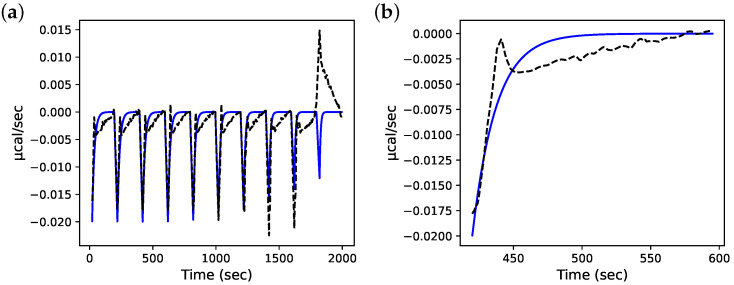
Estimation of PRDX1 dimer–decamer association and dissociation rate constants by fitting the ITC simulation model to experimental data in the top panel of Figure 1C in [7], which were digitised and processed as described in the text. The model injection interval (200 s) and total Prx in the simulated injections were as per the original experiment. The $k_{off}$ for decamer dissociation was estimated by fitting the model to these data; the solid line shows simulation results with the best-fit parameters. (**a**) (**—**) Model simulation with best-fit $k_{off}$ and and $k_{on}$ values (see text), (**- -**) digitised ITC data; (**b**) close-up view of a single ITC injection.

**Figure 5 antioxidants-12-01707-f005:**
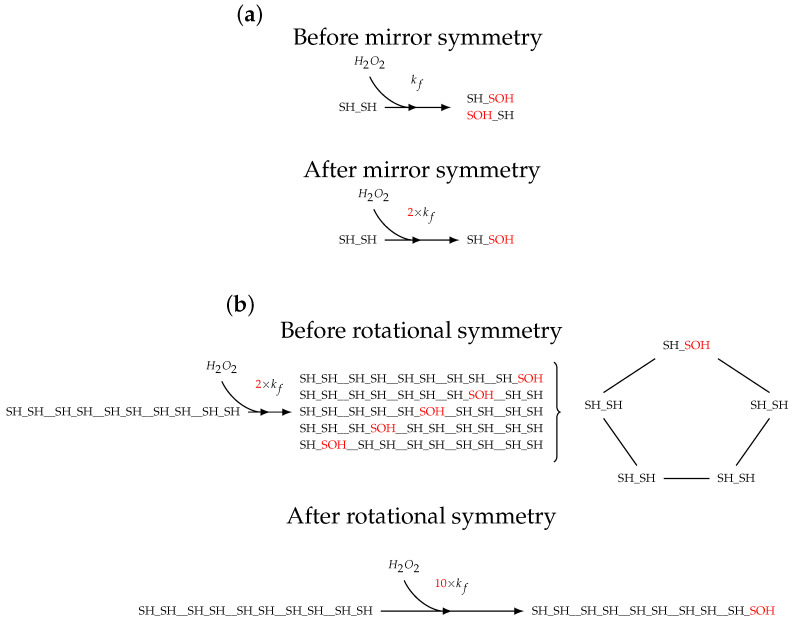
Scheme for scaling of rate constants by the amount of degeneracy (indicated by a red font colour) introduced from symmetry. (**a**) Degeneracy introduced by mirror symmetry within Prx dimers. Oxidation of one site in a reduced Prx dimer results in one of two products that are related by mirror symmetry. Therefore, the reactions can be collated into one reaction and the rate constant multiplied by a statistical factor of 2. This logic can be applied to the other reactions of Prx dimers. (**b**) Degeneracy introduced by rotational symmetry of Prx decamers. Taking mirror symmetry into account, oxidation of one site of a reduced Prx decamer results in 1 of 5 products that are related by rotational symmetry. Therefore, the reactions can be collated into one reaction with a statistical factor of 10. The same logic of symmetry can be applied to many of the Prx decamer species with other oxidation states, albeit resulting in less degeneracy.

**Figure 6 antioxidants-12-01707-f006:**
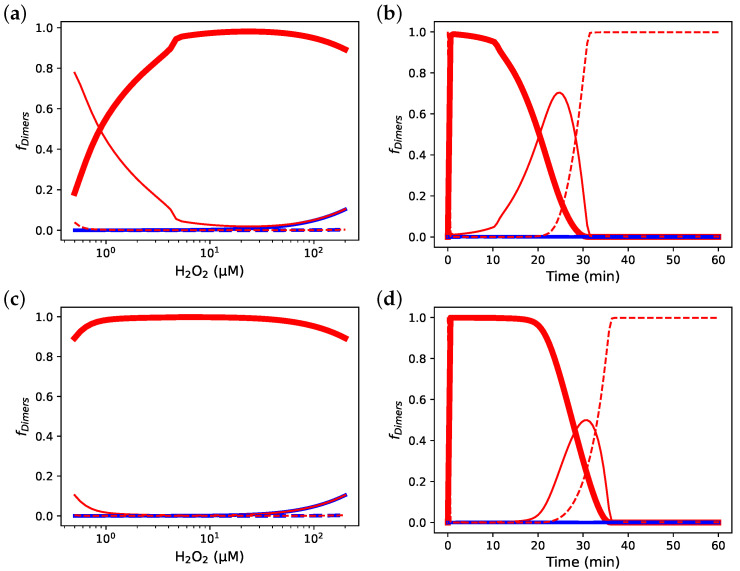
Incorporating decamerisation resolves a discrepancy in the response of PRDX2 oxidation state to hydrogen peroxide load between simulations and experimental evidence in the RBC. (**a**) Steady-state and (**b**) time-based simulations of a RBC PRDX2 model based on Model A in [14] with decamerisation added. (**c**,**e**) Steady-state and (**d**,**f**) time-based simulations of the original RBC PRDX2 models by Benfeitas et al. [14]: (**c**) and (**d**) Model A, (**e**) and (**f**) Model B. (- -) Fraction of reduced and sulfenilated Prx sites; (—) fraction of Prx dimers with a single disulphide bridge; (▬) fraction of Prx dimers with two disulphide bridges; (—) fraction of Prx dimers with two sulfinic sites; (- -) fraction of Prx dimers with a disulphide bridge and a sulfinic site.

**Figure 7 antioxidants-12-01707-f007:**
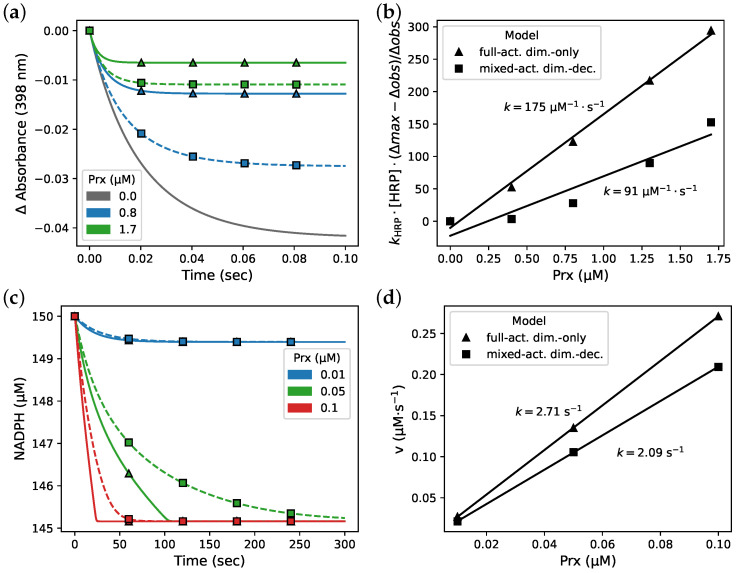
Incorporating Prx dimer–decamer topology creates an inhibition-like effect during simulated in vitro activity assays. Simulations of peroxidase assays with (Δ) the full-activity dimer-only and (□) the mixed-activity dimer–decamer models of PRDX2. (**a**) Simulation of a horse radish peroxidase (HRP) competitive assay with parameters to replicate the 0, 0.8, and 1.7 µM Prx traces from Figure 2 in Manta et al. [11] and (**b**) the associated determination of the Prx rate constant by fractional inhibition analysis using these traces as well as traces of simulated assays with 0.4 and 1.3 µM Prx. Solid lines, simulations with the full-activity dimer-only model. Dashed lines, simulations with the mixed-activity dimer–decamer model. (**c**) Simulation of a NADPH reduction assay with the Prx system including Trx and TRR, and (**d**) the associated Prx rate constant determination using the initial rate kinetics.

**Figure 8 antioxidants-12-01707-f008:**
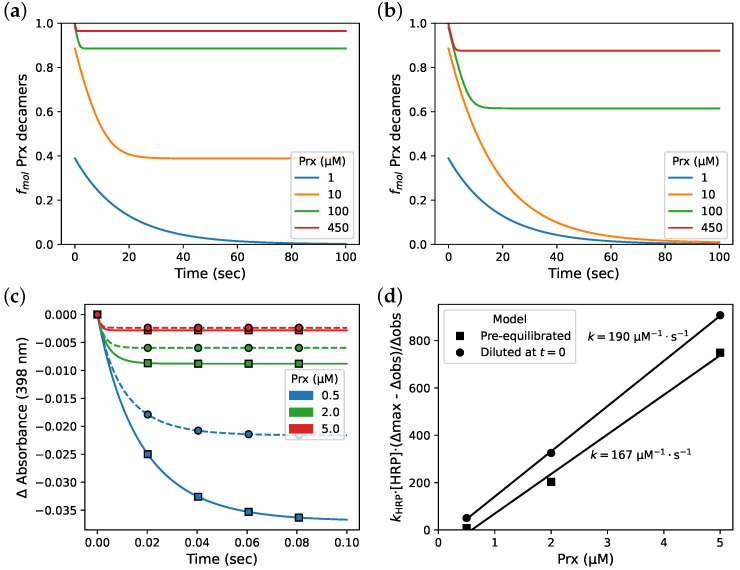
The time for a Prx solution to re-equilibrate after dilution is relevant to fast kinetic experiments. Simulation of (**a**) 10× and (**b**) 50× dilution of PRDX2 at equilibrium, starting at various initial concentrations. (**c**) Simulations of an HRP competition assay with the mixed-activity dimer–decamer model with (□—) assay substrates pre-equilibrated and (∘ - -) PRDX2 diluted 10× at $t_{0}$, and (**d**) the associated determination of the PRDX2 rate constant by fractional inhibition analysis.

## Data Availability

All data and code to reproduce the results presented in this manuscript are available from GitHub at https://github.com/Rohwer-Lab/Barry2023 (accessed on 25 August 2023).

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
