# Peer review of "Modelling the Decamerisation Cycle of PRDX1 and the Inhibition-like Effect on Its Peroxidase Activity"

_antioxidants, 2023, doi:10.3390/antiox12091707_

Round 1

Reviewer 1 Report

This manuscript takes existing data on decamer/dimer transformations of reduced Prdx1 to model the reaction and obtain association/dissociation rates for the reaction. The authors then incorporate this information into a published model of Prdx2 oxidation in the red blood cell. The original authors obtained a good fit only when they allowed for an (uncharacterised) inhibitor in the system, whereas the current manuscript shows good fit when decamer/dimer association is included in the model without requiring an inhibitor. This is an attractive explanation. The calculation of the of the association/dissociation rates seems reasonable, and the values obtained are a useful for increasing our understanding or Prdx kinetics. However I do have a number of queries about the rationale for this.

Major comments

 1.       I am not an expert on the details of kinetic modelling, but I do understand the principles involved and the general approach. In this case it is not clear to me (and I apologise if I have missed the detail) what the rationale is for including the dissociation in the red cell data. With low micromolar concentrations required for dissociation of the reduced Prdx and approximately 200 uM concentration in the red cell, dissociation seems unlikely. So does the model considers dissociation of the oxidised form – but what parameters are used? We know (for Prdx2) that it dissociates more readily, but several studies have shown that a major proportion is decameric (eg half at 10 uM, pH 7.4). I am not aware of any kinetic values, so please clarify what were used. Also, I presume that the model requires different reactivity of the decamer and dimer, either in the oxidation or reduction step – what evidence was used for this? I would like to see a much clearer explanation of how these factors were dealt with and more comment on the limitations of the model.

 2.       The manuscript uses data, particularly from refs 7, 11 and 14, for the modelling. Although for some detail it may be necessary to refer to the original publications, this manuscript should be understandable without having to do this. In many places (eg Figs 3&4) it is not possible to follow without going to the original. More explanation of the experimental setup is needed and in some cases it needs to be stated which specific figure (or part of figure) is referred to.

 3.       Fig 6. Explain what aspect of including dimer/decamer association is responsible for the fit.

 4.       Fig 7. Decamer/dimer transformation is a relevant consideration in experiments such as these as the reduced Prdx concentration is close to Kd and the dimer form of the disulfide would be favoured. However, more explanation of how this was treated is needed. It is difficult to decipher which curves from ref 11 are used. I presume from line 310 that a key feature of the modelling is lower peroxide reactivity (or peroxidase activity) of the dimer cf decamer. Please elaborate on this for Prdx 1 and the data used.

Minor points.

11.      Clarify for each figure which Prdx is being modelled.

22.     Figure 6. Note that data used in original simulation were from ref 12.

33.  Line 411. Is there evidence that 2-Cys Prdxs with 1 disulfide can’t form decamers?

Reviewer 2 Report

In the manuscript titled “Modelling the decamerisation cycle of PRDX1 and the inhibition-like effect on its peroxidase activity” the authors studied in detail the peroxiredoxin dimer-to-decamer transition and the inhibition-like effect on its peroxidase activity.

 I think this work is very important, it has been done with scientific rigour and is also very well written, in fact it is easily understood even by a reader who is not strictly from the field. I therefore accept the work after minor revision

My suggestion is

Define the limitations of this study in the discussion

Minor editing of English language required

Reviewer 3 Report

Peroxiredoxins are ubiquitous important enzymes that detoxify the reactive oxygen species. A dimer to decamer transition alters the enzymatic activity by 100-fold. However, there is no theoretical framework for this effect. In the present article, the authors have performed a theoretical study to explain this observation. Their model can be applied for the dimer to decamer transition for both PRDX1 and PRDX2, enzymes that have 77% identity. Further, they have performed various theoretical experiments (ITC, HRP competition etc) that explain their outcomes. The have found that the model simulated the HRP competition and the NADPH oxidation assays.

Specifically, their approach outlines the generation of a model that uses a reduced number of reactions and species that are necessary to model the decamer oxidation cycle. The authors show that dimers are the main form of Prx below the CTC but decamers dominate above CTC. Further, they show that a small number of decamers are present below CTC, while above CTC, the dimer concentration is almost constant.

Then the authors have conducted an in silico ITC experiment with improved parameters taking place and found that it matched the experimental data.

The also improved the model of PRXD2 in red blood cells by incorporating the decamerization reaction. This approximated the experimental data and avoided the use of the previously described inhibited PRX form that had not any evidence for its existence.

Finally, the authors conducted in silicon redox reactions that showed that decamerization reduced the overall activity of the PRX.

In conclusion, I recommend the publication of this article, since it is an in silico study the effectively recapitulates experimental data and provides explanation on the kinetics of PRX.

Round 2

Reviewer 1 Report

I am satisfied that the authors have addressed my concerns.

Author Response

We are happy to hear that the reviewer is satisfied that their concerns have been satisfactorily addressed.